

# Interactive Biogenic Emissions and Drought Stress Effects on Atmospheric Composition in NASA GISS ModelE

Elizabeth Klovenski[1], Yuxuan Wang[1], Susanne E. Bauer[2], Kostas Tsigaridis[2,3], Greg Faluvegi[2,3], Igor Aleinov[2,3], Nancy Y. Kiang[2], Alex Guenther[4], Xiaoyan Jiang[4], Wei Li[1], Nan Lin[5]

[1] Department of Earth and Atmospheric Sciences, University of Houston, Houston, TX, USA

[2] NASA Goddard Institute for Space Studies, New York, NY, USA

[3] Center for Climate Systems Research, Columbia University, New York, NYC, USA

[4] Department of Earth System Science, University of California – Irvine, Irvine, CA, USA

[5] Ministry of Education Key Laboratory for Earth System Modeling, Department of Earth System Science, Tsinghua University, Beijing, China

*Corresponding author:* Yuxuan Wang (ywang246@central.uh.edu)

**Key Points:**

- A new method to capture regional changes of isoprene drought stress is implemented for global usage in NASA GISS ModelE and is evaluated at the MOFLUX Ameriflux site located in Missouri.
- The inclusion of isoprene drought stress from 2003-2013 leads to a ~2.7% reduction in global decadal average of isoprene emissions in ModelE with up to ~20% reduction in drought-stricken regions.
- The model-tuned parameterization of isoprene drought stress reduces the overestimation of $\Omega$HCHO in the southeastern U.S and improves simulated $O_3$ during drought periods.

**Abstract.** Drought is a hydroclimatic extreme that causes perturbations to the terrestrial biosphere, and acts as a stressor on vegetation, affecting emissions patterns. During severe drought, isoprene emissions are reduced. In this paper, we focus on capturing this reduction signal by implementing a new percentile isoprene drought stress ($y_d$) algorithm in NASA GISS ModelE based on the MEGAN3 (Model of Emissions of Gases and Aerosols from Nature Version 3) approach as a function of a photosynthetic parameter ($V_{c,max}$) and water stress ($\beta$). Four global transient simulations from 2003-2013 are used to demonstrate the effect without $y_d$ (Default_ModelE) and with online $y_d$ (DroughtStress_ModelE). DroughtStress_ModelE is evaluated against the observed isoprene measurements at the Missouri Ozarks Ameriflux (MOFLUX) site during the 2012 severe drought where improvements in correlation coefficient indicate it is a suitable drought stress parameterization to capture the reduction signal during severe drought. The application of $y_d$ globally leads to a decadal average reduction of ~2.7% which is equivalent to ~14.6 Tg yr$^{-1}$ of isoprene. The changes have larger impacts in regions such as the Southeast U.S.. DroughtStress_ModelE is validated using satellite $\Omega$HCHO column from the Ozone Monitoring Instrument (OMI) and surface $O_3$ observations across regions of the U.S. to examine the effect of drought on atmospheric composition. It was found the inclusion of isoprene drought stress reduced the overestimation of $\Omega$HCHO in Default_ModelE during the



2007 and 2011 southeastern U.S. droughts and lead to improvements in simulated $O_3$ during
drought periods. We conclude that isoprene drought stress should be tuned on a model-by-model
basis, because the variables used in the parameterization responses are relative to the land
surface model hydrology scheme (LSM) and the effects of $y_d$ application could be larger than
seen here due to ModelE not having large biases of isoprene during severe drought.

**Plain Language Summary**: Severe drought stresses vegetation and causes reduced emission of
isoprene. We study the impact of including a new isoprene drought stress ($y_d$) parameterization
into NASA GISS ModelE called (DroughtStress_ModelE), which is specifically tuned for
ModelE. Inclusion of $y_d$ leads to better simulated isoprene emissions at the MOFLUX site
during the severe drought of 2012, reduced overestimation of OMI satellite $\Omega$HCHO
(formaldehyde column) and improved simulated $O_3$ (ozone) during drought.

## 1. Introduction

In present day conditions terrestrial ecosystems release about 1000 Tg C yr$^{-1}$ of biogenic
volatile organic compounds (BVOCs) into the atmosphere and there is an additional smaller
emission from marine ecosystems (Guenther *et al.* 2012). The majority of BVOCs emitted from
vegetation are isoprene and monoterpenes (Guenther *et al.* 2006; Guenther *et al.* 2012).
Representing over half of emitted BVOCs, isoprene is the dominant species globally with
reported ranges of 440-600 Tg C yr$^{-1}$ (Guenther *et al.* 2012) with high emission factors from
some, but not all, broadleaf trees including species of oak, willow, palm oil, and eucalyptus
(Benjamin *et al.* 1996; Geron *et al.* 2000). Isoprene is produced from carbon substrates generated
during photosynthesis and contributes to abiotic stress tolerance from water and temperature
stress (Loreto and Sharkey 1990; Monson et al. 2021). Isoprene emissions peak during warm,
sunnier months of the growing season (MAR-OCT) (Opacka *et al.* 2021). Isoprene has a
chemical lifetime of approximately one hour via oxidation by the hydroxyl radical (OH),
producing organic aerosols and oxidation products that contribute to ozone ($O_3$) formation
(Carlton *et al.* 2009). Biogenic isoprene emissions affect atmospheric composition and climate,
and in turn depend on drivers including light, temperature, photosynthetically active radiation
(PAR), leaf area index (LAI), water stress, ambient $O_3$, and $CO_2$ concentrations. Climate change-
related higher temperatures and $CO_2$ concentrations are separately expected to increase
emissions of BVOCs, which will impact tropospheric ozone and secondary organic aerosols
(SOA) formation. Increasing SOA will have a negative climate forcing effect through increased
scattering of sunlight, causing an aerosol direct forcing, and increased cloud condensation nuclei
(CCN), causing aerosol indirect forcing effects (Twomey 1974; Sporre *et al.* 2019). The
consideration of drought effects on BVOC emissions, as investigated in this study, will
counterbalance these effects, due to isoprene reductions caused by drought stress. During
drought, increases in SOA and $O_3$ are to be expected (Wang *et al.* 2017; Zhao *et al.* 2019), and
with isoprene reductions we expect a reduction in the magnitude of increase of both pollutants.
SOA acts as negative radiative forcing under future temperature and $CO_2$ increases (Zhu *et al.*
2017) and tropospheric $O_3$ and total $O_3$ acts as a positive radiative forcing (Skeie *et al.* 2020).




Drought is a common abiotic stress to terrestrial ecosystems characterized by low soil
moisture, usually associated with high temperature and low precipitation. However, even boreal
forests undergo winter drought due to frozen soils. Recent work has shown a strong correlation
between drought severity and fine-mode aerosols in the U.S. and estimated that regions
undergoing severe drought see up to 17% surface enhancement of aerosols during the growing
season (Wang *et al.* 2017). This suggests a strong perturbation of drought to atmospheric
aerosols, likely caused by changing BVOC emissions due to drought stress. Limited field and lab
measurements have shown that during drought, isoprene has a unique emission response where
initial increase in temperature causes an increase in emission, but prolonged or severe drought
causes a decrease of emissions due to the shutdown of physiological processes (Potosnak *et al.*
2014). This behavior is not reproduced by commonly used BVOC emission models such as the
Model of Emissions of Gases and Aerosols from Nature Version 2.1 (MEGAN2.1), which has a
simple drought algorithm which is often not used due to the unavailability of the required driving
variables in chemistry climate models (CCMs), and the Biogenic Emission Inventory System
(BEIS), which does not include a drought algorithm as an option.

Isoprene flux observations at the Missouri Ozarks (MOFLUX) Ameriflux site in Missouri (SI
Fig. S1) recorded a moderate drought in summer 2011 (Potosnak *et al.* 2014) and a particularly
severe drought event in summer 2012 (Seco *et al.* 2015). To the best of our knowledge, these are
the only in situ isoprene flux measurements capturing a drought anywhere. Using the MOFLUX
observations, Jiang *et al.* (2018) developed an isoprene drought stress activity factor for
MEGAN3 (Model of Emissions of Gases and Aerosols from Nature Version 3) designed to
reduce emissions of isoprene during drought. The previous MEGAN2.1 isoprene drought
parameterization utilized soil moisture and soil wilting point threshold to include impacts of
drought on photosynthetic processes. The MEGAN3 isoprene drought stress activity factor is a
more process-based parameterization based on a photosynthetic parameter ($V_{c,max}$) and water
stress ($\beta$) from the Community Land Model (CLM) as coupled with the CAM-Chem climate
model (Jiang *et al.* 2018). $V_{c,max}$ is the maximum carboxylation capacity of a leaf (usually in units
of micromole $CO_2$ per leaf area per time); that is, it is the ability of a plant to convert $CO_2$ into
sugar, and hence determine productivity of carbon substrates for biogenic volatile organic
compounds (BVOCs) production when no other conditions are limiting. $\beta$ is a scaling factor
between zero to one, used in CLM to reduce $V_{c,max}$ due to plant water stress. MEGAN3 isoprene
drought stress was also incorporated into the CSIRO chemical transport model (C-CTM) with
Australian land surface models Mk3.6 Global Climate Model and the Soil-Litter-Iso model with
a focus on Australia (Emmerson *et al.* 2019). Both prior modeling studies (Jiang *et al.* 2018;
Emmerson *et al.* 2019) only looked at the drought effects on $O_3$; here we study the combined
effect of drought on $O_3$ and formaldehyde column.





The accurate simulation of stress-affected emissions of isoprene during extreme hydroclimate
events (i.e. drought) is crucial to understanding vegetation-climate-chemistry feedbacks, because
isoprene is a precursor to tropospheric $O_3$ and SOA, both being climate forcers as well as air
pollutants. Here we focus on deriving a model-specific tuned isoprene drought stress factor that
is coupled into the existing MEGAN2.1 framework in NASA GISS ModelE, an Earth System
Model, to model the effect of drought on isoprene emissions and their effect on atmospheric
composition. The model-specific tuning is required due to different land system models
parameterizing key variables of $V_{c,max}$ and $\beta$ in different ways with varying distributions. The
model's drought effects will be extensively evaluated over the US, due to the availability of
observational evidence during drought (Wang *et al.* 2017). While the MOFLUX data are the only
available measurements of isoprene emissions during drought, formaldehyde (HCHO), the high
yield oxidation product of isoprene, can be used as a proxy for isoprene emissions (Zhu *et al.*
2016). **Section 2** describes the modelling approaches used to represent drought impacts on
isoprene emissions. **Section 3** describes the comparison of modeled isoprene emissions to
observations at the MOFLUX site during drought along with necessity of building a model
specific isoprene drought stress parameterization. **Section 4** details the comparisons between
simulation with model specific tuned isoprene drought stress (DroughtStress_ModelE) and
observational $O_3$, $PM_{2.5}$ (particulate matter $\leq 2.5\ \mu$m), and tropospheric formaldehyde columns
($\Omega$HCHO) over North America.
**2. Methods and Data**
**2.1. The biogenic emission model MEGAN**
MEGAN is a widely used BVOC emissions model that is implemented in many CCMs. Here
we describe briefly MEGAN2.1 as implemented in ModelE. MEGAN2.1 calculates the net
primary emissions for 20 compound classes, which are speciated into over 150 species such as
isoprene, monoterpenes, etc. (Guenther *et al.* 2012). The emissions rate (µg grid cell$^{-1}$ h$^{-1}$) of
each compound into the above canopy atmosphere from a model grid cell is calculated:
$Emission = EF \times y \times S$ (1)
where $EF$ (µg m$^{-2}$ h$^{-1}$) is emission factor per compound, $y$ is the dimensionless emission activity
factor that accounts for emission response to phenological and meteorological conditions, and S
is the grid cell area (m$^2$).
The emission activity factor $y$ for each compound is calculated following the MEGAN2.1
parameterization (Guenther *et al.* 2006; Guenther *et al.* 2012; Henrot *et al.* 2017).
$y = y_{CE} \times y_A \times y_d \times y_{Co_2}$ (2)





Where $y_{CE}$ is the canopy environment coefficient, assigned a value of one for standard
conditions, and it takes into account variations associated with LAI (m$^2$ m$^{-2}$), photosynthetic
photon flux density (PPFD) (μmol of photons in 400-700 nm range m$^{-2}$ s$^{-1}$), and temperature (K).
$y_A$ is the leaf age emission activity factor, parameterization of which is based on coefficients of
the decomposition of the canopy into new, growing, mature, and senescing leaves for current and
previous months' LAI (Guenther *et al.* 2006; Guenther *et al.* 2012). $y_d$ is the isoprene drought
stress activity factor and $y_{Co_2}$ is the isoprene emission activity factor associated with CO$_2$
inhibition (for all other compounds $y_d$ and $y_{Co_2}$ = 1). The biogenic emission module implemented
in ModelE follows the ECHAM6-HAMMOZ online MEGAN2.1 implementation (Henrot *et al.*
2017) in a CCM. Within ModelE the MEGAN2.1 module maps the 16 plant functional types
(PFTs) from Ent TBM (Terrestrial Biosphere Model) (Kim *et al.* 2015) into 16 MEGAN PFTs,
and contains 13 chemical compound classes. ModelE uses a modified MEGAN2.1 following
(Henrot *et al.* 2017) to provide a framework to simulate isoprene emissions, and uses prescribed
emissions factors per PFT to simulate emissions per compound class.

In Henrot *et al.* (2017) to avoid using a detailed canopy environment model calculating light
and temperature at each canopy depth, the Parameterized Canopy Environmental Emission
Activity (PCEEA) approach from Guenther *et al.* (2006) is used to replace $y_{CE}$ with a
parameterized canopy environment activity factor ($y_{LAI} \times y_P \times y_T$). With this approach the light
dependent and light independent factors are multiplied by $y_{LAI}$ not LAI so they are not directly
proportional to LAI. This approach allows for calculation of light dependent emissions following
isoprene emission response to temperature, where its assumed the light dependent factor (LDF)
equals one for isoprene and light independent emissions follow the monoterpene exponential
temperature response. Please see Guenther *et al.* (2006); Guenther *et al.* (2012); Henrot *et al.*
(2017) for activity factor parameterizations. At any given time step in ModelE, the emissions
formula for a compound class (c) and PFT (i), in units of kg m$^{-2}$ s$^{-1}$ is given by:

$Emission_{i,c} = (1x10^{-9}/3600) \times (EF_{i,c} \times PFTboxf_i) \times y_{LAI} \times y_A \times y_d \times y_{co_2} \times ((1 -$
$LDF) \times y_{TLI} + LDF \times y_P \times y_{TLD}) \times SF_c \times MWC_c$  (3)

where EF$_{i,c}$ is the emissions factor (μg m$^{-2}$ hr$^{-1}$) for a given PFT and compound class, PFTboxf$_i$ is
the fraction of the grid cell (ranging from zero to one) covered by PFT *i*, and SF$_c$ is a linear scale
factor for compound class c. The activity factors, y, listed in Equation (3) are unitless and
account for the emissions response to leaf area index (LAI), aging (A), drought (d), CO$_2$ (CO$_2$),
and PPFD (P). The LDF, weights the contributions from light independent ($y_{TLI}$) and light
dependent ($y_{TLD}$) emissions response to temperature. MWC$_c$ stands for a molecular weight
conversion to remove non-carbon mass, if appropriate. (1x10$^{-9}$/3600) is a timestep conversion for
seconds in an hour. Note that although the drought activity factor $y_d$ is present in ModelE, it is
set to equal one in all cases prior to this work, meaning no drought effects on BVOC emissions
in the model.





For example, the emission formula for the compound class of isoprene in ModelE for PFT i is as follows (where LDF=1):

$$Isoprene_i = (1x10^{-9}/3600) \times (EF_{i,isoprene} \times PFTboxf_i) \times y_{LAI} \times y_A \times y_d \times y_{co_2} \times (y_P \times y_{TLD}) \times SF_{isoprene} \times (60.05/68.12) \tag{4}$$

## 2.2  MEGAN2.1 Isoprene Drought Stress Emission Algorithm

Guenther *et al.* (2006) introduced isoprene drought stress as a soil moisture dependent algorithm called $y_{SM}$. This isoprene drought stress activity factor relied upon soil moisture and wilting point to apply drought stress to isoprene emissions. The algorithm for soil moisture isoprene drought stress is as follows:

$$y_{SM} = 1 \ when \ \theta > \theta_1 \tag{5a}$$
$$y_{SM} = \frac{\theta - \theta_w}{\Delta_{\theta_1}} \ when \ \theta_w < \theta < \theta_1 \tag{5b}$$
$$y_{SM} = 0 \ when \ \theta < \theta_w \tag{5c}$$

where $\theta$ is soil moisture (volumetric water content m$^3$ m$^{-3}$), $\theta_w$ is the point beyond which plants cannot extract water from soil, known as the wilting point, m$^3$ m$^{-3}$, $\Delta_{\theta_1}$ (=0.06 in Guenther et al. 2006 and =0.04 in Guenther et al. 2012) is an empirical parameter, and $\theta_1$ is defined as $\theta_w + \Delta_{\theta_1}$. Soil moisture and wilting point are not widely available parameters in models, and $y_{SM}$ was not widely adopted to represent isoprene drought stress as studies showed substantial uncertainty associated with soil moisture predicted response of isoprene emission to water stress and in selection of wilting point values (Müller *et al.* 2008; Tawfik *et al.* 2012; Sindelarova *et al.* 2014; Huang *et al.* 2015; Jiang *et al.* 2018). There also exist challenges associated with validating soil moisture datasets due to the limited spatial coverage of in-situ root-zone measurements in the contiguous United States (Ochsner *et al.* 2013). A study found that the accurate simulation of soil moisture in land surface models was highly model-dependent, due to the differing horizontal and vertical spatial resolution of such models at large scales (Koster *et al.* 2009). Potosnak *et al.* (2014) determined that the selection of different wilting point values greatly impacted the drought impacts on biogenic isoprene emission. With these associated challenges, it was rare to find isoprene drought stress implemented in CCMs, thus a new isoprene drought activity factor needed to be developed that could be easily incorporated into a variety of models that had a land surface model (LSM) or terrestrial biosphere model (TBM).

## 2.3  MEGAN3 Isoprene Drought Stress Emission Algorithm

Jiang *et al.* (2018) developed a new isoprene drought stress activity factor in MEGAN3 that focuses on photosynthetic carboxylation capacity and water stress to model reductions of vegetative isoprene during drought. The algorithm was developed using isoprene flux observations during the severe drought of the summer of 2012 and less severe drought of 2011 (Potosnak *et al.* 2014; Seco *et al.* 2015) at MOFLUX. The MOFLUX site is located in the University of Missouri Baskett Wildlife Research area in central Missouri which is known as the isoprene volcano (Wells *et al.* 2020). The MOFLUX site is comprised primarily of deciduous



broadleaf trees, primarily oaks, known to emit high quantities of isoprene. All meteorological
data from the site comes from the Ameriflux website (https://ameriflux.lbl.gov/sites/siteinfo/US-
MOz#overview).

We refer to the original MEGAN3 drought stress developed by Jiang *et al.* (2018) to be
**DroughtStress_MEGAN3_Jiang**, and the corresponding parameterization for isoprene activity
factor during drought where $(y_d)$ is a function of PFT and where the values of $V_{cmax}$ and $\beta$ are
specified by PFT is:

$y_d = 1$ , $when\ \beta\ \geq 0.6$                                                                          (6a)
$y_d = \frac{(V_{c,max} \times \beta)}{\alpha}$ , $when\ \beta < 0.6, \alpha = 37$                           (6b)
$0 \leq y_d \leq 1$                                                                                          (6c)

$Isoprene_i = (1x10^{-9}/3600) \times \left( EF_{i,isoprene} \times PFTboxf_i \right) \times y_{LAI} \times y_A \times y_d \times y_{co_2} \times (y_P \times y_{TLD}) \times$
$SF_{isoprene}$                                                                                             (7)

The drought stress activity factor, $y_d$, in DroughtStress_MEGAN3_Jiang was originally
developed using the Community Land Model Version 4.5 (CLM4.5) (Jiang *et al*. 2018). The
photosynthetic parameter used is $V_{c,max}$, which is the maximum rate of leaf-level carboxylation.
In ModelE, $V_{c,max}$ is scaled with an enzymatic kinetics response to temperature, and drought
stress reduces leaf stomatal conductance, thereby reducing photosynthetic activity through $CO_2$
diffusion limitation rather than by reduction of $V_{c,max}$. In CLM4.5, $V_{c,max}$ is a function of nitrogen
(Jiang *et al.* 2018). Water stress in CLM4.5 is based on soil texture (Clapp and Hornberger
1978), and it is a function of soil water potential of each soil layer, wilting factor, and PFT root
distribution. Water stress $(\beta)$ ranges from zero when a plant is completely stressed to one when a
plant is not undergoing stress. In CLM4.5, $V_{c,max}$ is scaled online by $\beta$ before being applied into
the isoprene drought activity parameterization, thus this scaling step is not reflected in the
equations shown by Jiang *et al.* (2018). Since ModelE does not scale $V_{c,max}$ by $\beta$ (instead,
ModelE scales leaf stomatal conductance by $\beta$), to reproduce the original scheme by Jiang *et al.*
(2018) as much as possible in ModelE, we scaled $V_{c,max}$ with $\beta$ inside the equation of isoprene
drought activity factor as in Eq. (6b). $y_d$ as defined in Eq. (6) is then applied in ModelE as an
activity factor into the MEGAN2.1 isoprene emissions equation per every plant functional type
(PFT) and the modeling results from this simulation are referred to as
**DroughtStress_MEGAN3_Jiang**. The $y_d$ ranges from zero to one and is designed to reduce
isoprene emissions during severe and prolonged drought.

**2.4  NASA GISS ModelE Climate Chemistry Model**
NASA GISS ModelE2.1 is an Earth System Model (ESM) with a horizontal and vertical
resolution of $2°$ degrees in latitude and $2.5°$ degrees in longitude with 40 vertical layers from the
surface to 0.1 hPa (Kelley *et al.* 2020). The climate model is configured in CMIP6 (Coupled



Model Intercomparison Project Phase 6) configuration (Miller *et al.* 2021) with fully coupled
atmospheric composition with interactive gas-phase chemistry. The model described here is
driven by historical Atmospheric Model Intercomparison Project simulations (AMIP), using
prescribed ocean temperature and sea ice datasets. There are two aerosol schemes to choose
from: MATRIX ("Multiconfiguration Aerosol TRacker of mIXing state") (Bauer *et al.* 2008) a
microphysical aerosol scheme and OMA (One-Moment Aerosol) mass-based aerosol scheme
(Koch *et al.* 2006; Miller *et al.* 2006; Bauer *et al.* 2007; Tsigaridis *et al.* 2013; Bauer *et al.* 2020).
Here we use the OMA scheme, due to its better representation of secondary organic aerosol
chemistry (Tsigaridis *et al.* 2013). SOA is calculated using the CBM4 chemical mechanism to
describe the gas phase tropospheric chemistry together with all main aerosol components
including SOA formation and nitrate, and is calculated using four tracers in the model. Isoprene
(VOCs) contribute to the formation of SOA. OMA has 34 tracers for the representation of
aerosols that are externally mixed, except for mineral dust that can be coated (Bauer *et al.* 2007),
and has prescribed constant size distribution (Bauer *et al.* 2020). OMA aerosol schemes are
coupled to the stratospheric and tropospheric chemistry scheme (Shindell *et al.* 2013) which
includes inorganic chemistry of $O_x$, $NO_x$, $HO_x$, CO, and organic chemistry of $CH_4$ and higher
hydrocarbons, with explicit treatment of secondary OA (organic aerosol), and the stratospheric
chemistry scheme which includes chlorine and bromine chemistry together with polar
stratospheric clouds. $O_3$ and aerosols impact climate via coupling to the radiation scheme, and
aerosols serve as cloud condensation nuclei (CCN) for cloud activation. The model includes the
first indirect effect. Sea salt, dimethyl sulfide (DMS), and biogenic dust emission fluxes are
calculated interactively, while anthropogenic dust is not represented in ModelE2.1. Other
anthropogenic fluxes are from the Community Emissions Data System Inventory (CEDS)
(Hoesly *et al.* 2018) and biomass burning is from GFED4s (Global Fire Emissions Database with
small fires) inventory (van Marle *et al.* 2017) for 1850-2014.
Vegetation activity in ModelE is simulated with a dynamic global vegetation model, the Ent
Terrestrial Biosphere Model (Ent TBM) (Kim *et al.* 2015). In standard ModelE experiments, the
Ent TBM prescribes satellite-derived vegetation canopy structure (plant functional type, canopy
height, monthly leaf area index) (Ito *et al.* 2020) as boundary conditions for coupling the
biophysics of canopy radiative transfer, photosynthesis, vegetation and soil respiration, and
transpiration with the land surface model and atmospheric model. These processes provide
surface fluxes of $CO_2$ and water vapor, and surface albedo is specified by cover type and season.
ModelE uses the MEGAN2.1 BVOC emissions model to simulate interactive biogenic emissions
from vegetation (Guenther *et al.* 2006; Guenther *et al.* 2012). Ent TBM water stress is calculated
as a scaling factor between zero and one as a function of relative extractable water (REW) for the
given soil texture and PFT-dependent levels of REW for onset of stress and wilting (Kim *et al.*
2015); this scaling has been updated since Kim *et al.* (2015) to be a function of the water stress
factor of only the wettest soil layer in the PFT's root zone. Ent TBM uses a leaf-level model of
coupled Farquhar-von Caemmerer photosynthesis/Ball-Berry stomatal conductance (Farquhar



and von Caemmerer 1982; Ball and Berry 1985). The model calculates an unstressed leaf
photosynthesis rate and stomatal conductance, then applies its water stress scaling factor to scale
down leaf stomatal conductance, to emulate how hormonal signaling by roots under water stress
induces stomatal closure. Since there is a coupling of transpiration and $CO_2$ uptake through
stomatal conductance, water stress thereby also reduces photosynthesis rate through the
limitation on $CO_2$ diffusion into the leaf; this is different from CLM4.5's approach, which
instead reduces $V_{c,max}$. Canopy radiative transfer in the Ent TBM scales leaf processes to the
canopy scale by calculating the vertical layering of incident photosynthetically active radiation
on sunlit versus shaded leaves. The different PFTs in Ent TBM have different critical soil
moisture values for the onset of stress (when stomatal closure begins in response to drying soils)
and their wilting point (when the plant is unable to withdraw moisture from the soil and complete
stomatal closure occurs). It should be noted that the GISS land surface model is wetter than
observed soil moisture (Kim *et al.* 2015). $V_{c,max}$ is a function of a $Q_{10}$ temperature function in
ModelE. Since nitrogen dynamics are not represented yet in the Ent TBM, leaf nitrogen is fixed
and therefore $V_{c,max}$ is not dynamic with nitrogen as in CLM4.5. The $Q_{10}$ coefficient is often used
to predict the impact of temperature increases on the rate of metabolic change (Rasmusson *et al.*
342  2019).

To emulate the MEGAN/CLM representation of drought stress, in this study, in the Ent TBM
leaf model, we applied a reduction in $V_{c,max}$ with water stress as shown in Eq. (6b). It is important
to note that the reduction of $V_{c,max}$ with water stress in Eq. (6b), is not used outside the isoprene
drought stress parameterization, so the $V_{c,max}$ reduction is not applied to the calculation of
photosynthetic $CO_2$ uptake; this avoids applying another secondary indirect scaling to
conductance, since the Ent TBM already applies its water stress factor to reduce stomatal
conductance.
For this study, ModelE2.1 was configured with a transient atmosphere and ocean using a
prescribed sea surface temperature (SST) and sea ice (SSI) according to observations. The
transient simulations contain continuously-varying greenhouse gases in order to represent a
realistic mode in present day. To facilitate direct comparison with atmospheric composition
observations as in this study, meteorology is nudged to the National Centers for Environmental
Prediction (NCEP) reanalysis winds. Four transient ModelE simulations were run for the period
of 2003-2013 with a three-year spin-up using MEGAN2.1 with varying configurations for
isoprene drought stress to be described below. The authors found that the default MEGAN
implementation in ModelE2.1 underestimates isoprene and monoterpene emissions, thus
appropriate scaling factors ($SF_c$) were applied to match literature for global annual emission
estimates, 1.8 for isoprene and 3 for monoterpenes to match literature estimates of around ~500
Tg C of isoprene and ~130 Tg C of monoterpenes (Arneth *et al.* 2008; Guenther *et al.* 2012).
**2.5  Observations of Isoprene Emissions at MOFLUX during Drought of 2011-2012**



The MOFLUX site located at 38.7441°N, -92.2000°W (latitude, longitude) is comprised
mostly of deciduous broadleaf forests dominated by oak-hickory forest and the climate is
classified as humid subtropical with no dry season and hot summers. The site experienced a mild
drought in the mid to late summer of 2011 and an extreme to exceptional drought from the mid
to late summer of 2012 when concurrent biogenic isoprene flux measurements were taken. The
2011 drought was not as severe as the drought of summer of 2012. The ecosystem response of
isoprene has two stages including a mild phase of drought stress where emissions are stimulated
by increases in leaf temperature due to reduced stomatal conductance while in the second stage
of drought, the more severe phase of drought stress, emissions are suppressed by reduction in
substrate availability or isoprene synthase production (Potosnak *et al.* 2014; Seco *et al.* 2015).
In 2011, the spring was wet but the drought started to appear in June due to lack of rainfall
while temperatures broke records and continued through July (Potosnak *et al.* 2014; Jiang *et al.*
2018). However, the USDM (U.S. Drought Monitor) did not capture this drought signal from
June - July and only showed abnormally dry periods from August 2 - August 16, and never went
into extreme (D2) or severe drought stage (D3). This suggests 2011 summer was a useful case
only for studying drought response of isoprene during weak drought conditions. The highest
observed isoprene fluxes were from July 11 – August 3 shown in Fig. 1a. Potosnak *et al.* (2014)
reported that from July 14 - August 10 their MEGAN2.1 simulations consistently underestimated
isoprene emissions during onset of drought and overestimated as drought progressed from
August 18 to September 2. From August 3 – August 23 there was a total of 65 mm of
precipitation, which led to an increase in observed soil moisture. It was suggested that since
observed soil moisture increases during the period of drought progression when isoprene is
decreasing (August 18 - September 2) relative to the onset of drought (July 14 - August 10), this
indicates the response to drought stress during this year is time dependent, and a time-
independent algorithm based on soil moisture will not capture the relevant processes during a
less severe drought year. It was also noted that MEGAN2.1 underpredicts during the cooler
months of May-June and underpredicts during the warmer month of July (Potosnak *et al.* 2014),
and only overpredicts during small portions of August-September as denoted by a grey box in
Fig. 1a. With this pattern of underprediction observed in MEGAN2.1 simulations and also seen
in Default_ModelE, as well as weak drought conditions as stated above, 2011 is not an ideal year
to tune an isoprene drought stress algorithm to target the reduction period caused by drought
stress.
In 2012, there were three unique periods that displayed the development of a severe drought
that make it ideal to tune an isoprene drought stress algorithm. Shown in Fig. 1b is the daily
averaged isoprene flux broken up into three periods. We define the MAXVOC episode from
May 1 - July 16, severe drought period (July 17-August 31) shaded in brown in Fig. 1b, and the
drought recovery period (September 1-31). Although Seco *et al.* (2015) defined MAXVOC from
June 18 – July 31, they identified July 16 as the transitional stage between MAXVOC episode



and severe drought. Thus, our work used July 16 to separate MAXVOC and severe drought
periods. The periods of pre-drought (prior to May 31) and mild drought identified by Seco *et al.*
(2015) from May 31- June 14 are included in the MAXVOC period, because during this time
period a typical seasonal pattern of increasing emissions with increasing temperatures is shown,
and there is no indication of decreasing emissions due to drought stress. The mild drought period
(May 31- June 14) corresponds to USDM periods of abnormally dry and moderate drought.
Isoprene emissions continue to increase during the beginning of summer, which is supported by
several studies that show isoprene emissions during the first stages of drought increase even
though there is a decrease in $CO_2$ fixation, which is attributed to drought induced stomatal
closure and rising leaf temperature and decreasing transpirational cooling and $CO_2$ concentration
in the leaf (Rosenstiel *et al.* 2003; Pegoraro *et al.* 2004; Potosnak *et al.* 2014; Seco *et al.* 2015).
Separating MAXVOC and severe drought period allows for the algorithm development to target
the latter severe drought stage where isoprene reduction occurs, while not reducing emissions
during the early, and less severe, stages of drought. During the severe drought period, total
annual precipitation was the lowest in a decade while soil water content reached its minimum at
the end of August when the drought peaked (Jiang *et al.* 2018). During the severe drought there
is a marked decrease in isoprene flux shown by the brown shaded box coinciding with lower $\beta$
values. It is well established that isoprene emissions are linked to high temperatures (Singsaas
and Sharkey 2000), and without the contributing factor of drought there should be a rising
increase in isoprene emissions in July and August. The severe drought period encompasses
periods of severe and extreme drought identified by the USDM. July 3 marks the first week
indicated by USDM of severe drought and July 31 marks the first week of extreme drought.
During severe drought isoprene production is suppressed by reductions in substrate availability
and isoprene synthase transcription (Potosnak *et al.* 2014). Rain events at the end of August led
to drought recovery and soil water content started to increase, which is indicated by increasing $\beta$
values shown in the drought recovery period indicated in purple in Fig. 1b. Overall, 2012 shows
a complete development of drought conditions that affect isoprene emissions and will provide
useful constraints on the drought stress factor parameterization: a MAXVOC period that
encompasses pre- and mild drought periods, a severe drought period (July 17 – August 31), and a
drought recovery period (September 1-30).






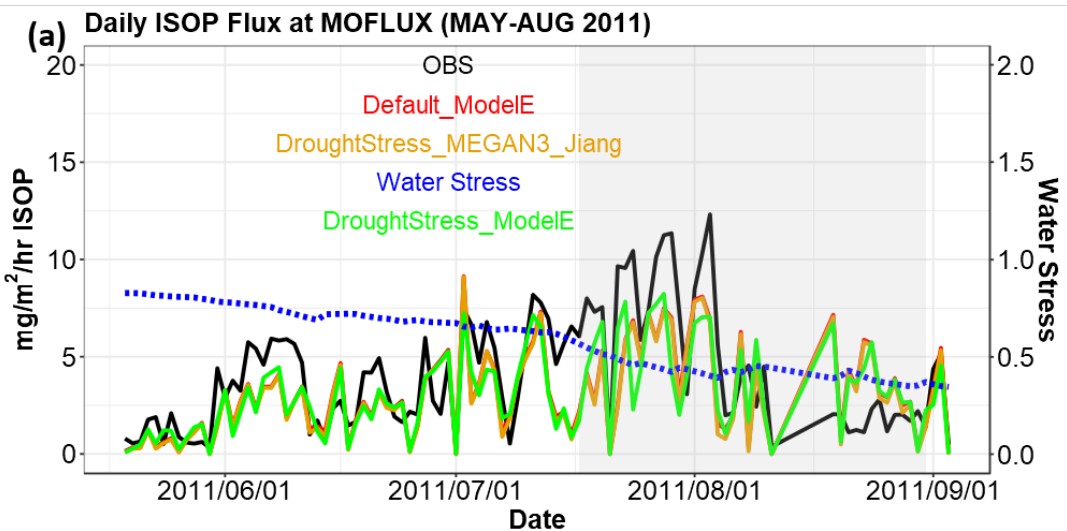

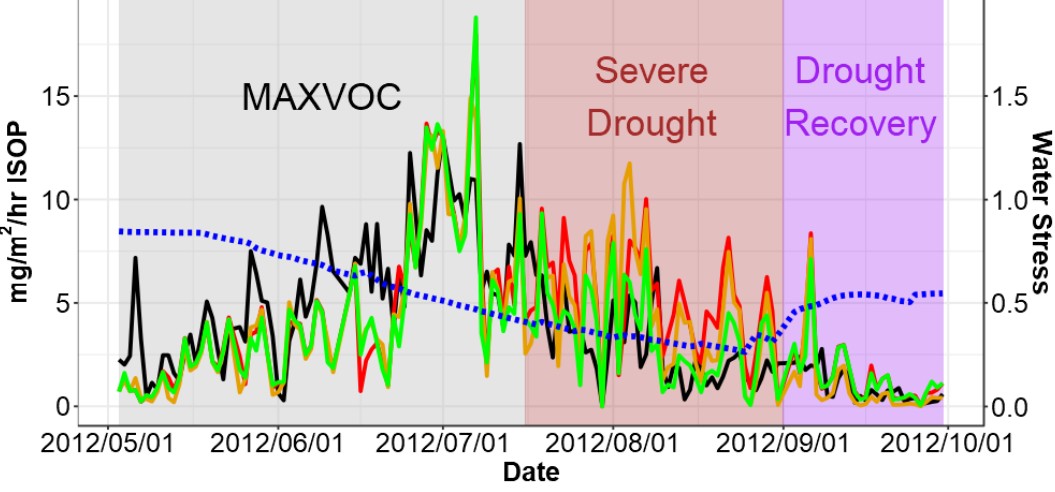

**Figure 1. Daily isoprene emissions flux at MOFLUX (MAY-AUG 2011 and MAY-SEP 2012) LST timeseries are shown.**
**Black shows observed isoprene emissions (abbreviated as ISOP), red shows Default_ModelE without isoprene drought**
**stress, orange shows DroughtStress_MEGAN3_Jiang, and green shows DroughtStress_ModelE with units of mg/m²/hr of**
**isoprene. (a) Shaded in the grey region from JUL 17 through AUG 31 of 2011, is the period where water stress falls below**
**0.4 for short periods. (b) Shaded in grey is the MAXVOC period, and shaded in brown is the period of severe/extreme**
**drought from July 17 through August 2012, and shaded in purple is the drought recovery period.**

**2.6 Offline Isoprene Emissions Model**



An offline model was created based on the isoprene emissions formula Eq. (4) of the
MEGAN module contained in ModelE in order to develop the new parametrization in a timely
fashion without waiting for online transient simulations to complete. ModelE was first run in a
default transient simulation with MEGAN2.1 where no isoprene drought stress was applied,
referred to as **Default_ModelE**, from which the MEGAN activity factors and variables required
to drive the offline calculation of isoprene emissions were output and archived. The offline
model was then driven by these outputs at the half hourly timestep to match with the 30-minute
timestep in the online calculation of physics and the MEGAN module. The offline model was
verified by making sure outputs of isoprene emissions matched the online Default_ModelE
simulation. With the verified offline model, different parameterizations of isoprene drought
stress could be tested and cross verified with observations at MOFLUX. The offline model is
used to derive a model specific $\alpha$ and $\beta$ threshold (Eq. (6a-6c)) for ModelE in order to create the
appropriate parameterization of a model specific isoprene drought stress in ModelE known as
**DroughtStress_ModelE**, described in Section 3.3. Since models calculate water stress and $V_{c,max}$
in different ways, the offline model is the necessary step to derive model-specific water stress
thresholds to target drought periods and ensure $\alpha$ and $\beta$ are being applied correctly.
**2.7   ModelE Sensitivity Simulations**
Four transient global ModelE simulations were configured for the period of 2003-2013 with
a three-year spin-up, as described in **Table 1**. A default simulation (Default_ModelE) that set $y_d$
=1 was performed where no isoprene drought stress parameterization was applied. A second
simulation named DroughtStress_MEGAN3_Jiang was performed as a sensitivity test to
determine the efficacy of the DroughtStress_MEGAN3_Jiang algorithm Eq. (6a-6c), which is
not tuned specifically for ModelE, and was originally developed by Jiang *et al.* (2018) as a non-
model specific tuned isoprene drought stress formula to be used widely in models. A third
simulation was performed with the offline derived ModelE tuned isoprene drought stress
parameterization to best fit MOFLUX observations (MOFLUX_DroughtStress) using Eq. (8a-
8c) to be described in Section 3.2. A fourth simulation called DroughtStress_ModelE was
performed using a subset of parameters derived from MOFLUX_DroughtStress but a different
drought activation method in Section 3.3 using Eq. (10a-10b).





**Table 1. ModelE Online Transient Simulation Descriptions**

| Simulation Name | Drought Stress | Isoprene Emission Eqn. | $\beta$ Threshold | $\alpha$ |
|---|---|---|---|---|
| 1)  Default_ModelE | NO | Eq. (4) | N/A | N/A |
| 2)  DroughtStress_MEGAN3_Jiang | YES  Eq. (6a-6c) | Eq. (7) | $\beta < 0.6$ | 37 |
| 3)  MOFLUX_DroughtStress | YES  Eq. (8a-8c) | Eq. (9) | $0.25 < \beta <$ 0.40 | 100 |
| 4)  DroughtStress_ModelE | YES  Eq. (10a-10b) | Eq. (9) | $\beta < 4^{th}$ percentile | 100 |


**3. Development of Model specific Drought Stress Parameterization**

488         **3.1. MOFLUX Single Site Observational Comparison to Model**

489         Shown in Fig. 1a is the 2011 timeseries of biogenic isoprene flux at the MOFLUX site of two

online simulations Default_ModelE (red) and DroughtStress_MEGAN3_Jiang (orange)
compared to observations (black). In 2011, Default_ModelE tended to underestimate isoprene
flux during onset of drought (July 14 - August 10) and had minor periods of overestimation
during drought progression (August 18 – September 2) which was also seen by MEGAN2.1
simulations of Potosnak *et al.* (2014). DroughtStress_MEGAN3_Jiang simulation applied
isoprene drought stress from mid-July through September when $\beta$ fell below the 0.6 threshold
identified by Jiang et al. (2018). In the DroughtStress_MEGAN3_Jiang simulation it is shown
that during the drought progression stage, DroughtStress_MEGAN3_Jiang isoprene is reduced
compared to Default_ModelE, but reductions are not strong enough to align with lower observed
values for a majority of this period. The timeseries shows that there is little deviation between
the Default_ModelE and DroughtStress_MEGAN3_Jiang during the 2011 mild drought.

502         Shown in Fig. 1b is the 2012 timeseries of biogenic isoprene flux at the MOFLUX site of two

online simulations Default_ModelE and DroughtStress_MEGAN3_Jiang compared to
observations, with $\beta$ (blue). Default_ModelE typically underestimates isoprene flux during the
MAXVOC period, overestimates during the severe drought period, and reproduces the drought
recovery period sufficiently except for September 6 where the model greatly overestimates
leading to a peak not matched by observations. During the severe drought period the
Default_ModelE mean bias (MB) $\cong$2.20 mg/m$^2$/hr and the normalized mean bias (NMB) $\cong$
76.10%. $\beta$ daily average values fell below the 0.60 threshold on June 20 and continued below the
threshold through September 3. With the $\beta$ falling below 0.60, the
DroughtStress_MEGAN3_Jiang simulation starts reducing isoprene during the MAXVOC
period and continues to reduce through the drought recovery period. This leads to compounding
the underestimation during the MAXVOC period, small corrections to overestimation during
severe drought but missing the peak overestimations, and too large of reductions of isoprene
during drought recovery period. During the severe drought period the MB of



DroughtStress_MEGAN3_Jiang was $\cong 1.61$ mg/m$^2$/hr and the NMB was $\cong 55.81\%$.
DroughtStress_MEGAN3_Jiang thus decreased the overestimation by ~20.29% during the
severe drought period. The timeseries comparison for 2012 indicates the parameters in the Jiang
et al. parameterization resulted in only minor improvements in ModelE for the severe drought
period, because they were tuned for CLM4.5. The DroughtStress_MEGAN3_Jiang simulation
shows that the $\alpha$ and $\beta$ need to be tuned on a model-by-model basis. Based on these minor
improvements, and the differences in how $V_{c,max}$ and $\beta$ are calculated in CLM4.5 versus Ent
TBM, it was clear a model tuned parameterization could be used to further improve the
relationship of simulated isoprene emissions during drought.

**3.2 Site Tuned MOFLUX_DroughtStress Parameterization**
Using the offline isoprene emissions model (Section 2.6) driven by catalogued variables from
each time step of the **Default_ModelE** simulation and the MOFLUX biogenic isoprene flux
measurements for 2012, we describe here how a water stress threshold to target severe/extreme
drought periods and a model appropriate empirical variable ($\alpha$) were derived to create the
isoprene drought stress parameterization based upon the framework of Eq. (6a-6c), called
**MOFLUX_DroughtStress**. MOFLUX_DroughtStress was developed to target the 2012 severe
drought period shown in Fig. 1b as this period is when the model overestimates despite
observations showing decreasing emissions during drought. The water stress threshold range
targeting the severe drought period determines when the isoprene drought stress is applied and it
is bounded to exclude the period of drought recovery and the onset of drought when isoprene
emissions are still increasing. The range of $\beta$ specific to ModelE is 0.25 to 0.40 during the severe
drought period, which differs from the CLM4.5 threshold of 0.60 as it is a model specific
parameterization. Isoprene drought stress in MOFLUX_DroughtStress is thus applied only when
$\beta < 0.40$, and at all other $\beta$ values $y_d = 1$.

To find the empirical variable, $\alpha$, an offline sensitivity analysis was conducted using the
offline isoprene emissions model with 0.25 to 0.40 as the $\beta$ threshold to activate isoprene
drought stress. The PFT weighted value of $V_{c,max}$ and $\beta$ were used to calculate the $y_d$ in the
offline isoprene emissions model. A range of $\alpha$ values from 60 to 160 were tested in Eq. (8a-8c)
to find $y_d$. $y_d$ dependence on the value of $\alpha$ was fed into Eq. (9) to output offline isoprene
emissions. The offline modeled emissions from Eq. (9) were evaluated against observed isoprene
fluxes at MOFLUX, and it was determined that $\alpha = 100$ gave the best fit and strongest
relationship between the offline modeled emissions and measured isoprene at MOFLUX. The $\alpha$
variable, though empirically derived, is strongly related to the model specific $V_{c,max}$ which is why
our alpha differs from DroughtStress_MEGAN3_Jiang, where $\alpha = 37$. Based on the offline
emissions comparisons to observed it was determined that **MOFLUX_DroughtStress** is defined
as follows:

$y_d = 1 \ (\beta \geq 0.4)$                                                   (8a)



$y_d = \frac{(v_{c,max} \times \beta)}{\alpha}$ $(0.25 < \beta < 0.40)$ where $\alpha = 100$          (8b)
$y_d = 1$ $(\beta \leq 0.25)$          (8c)

$Isoprene_i = (1x10^{-9}/3600) \times (EF_{i,isoprene} \times PFTboxf_i) \times y_{LAI} \times y_A \times y_d \times y_{co_2} \times (y_P \times y_{TLD}) \times$
$SF_{isoprene}$          (9)

Where $y_d$ uses the area weighted average over PFTs of $v_{cmax}$ and $\beta$ in Eq. (8a-c), and thus $y_d$ in
Eq. (9) is not a function of PFT, which differs from DroughtStress_MEGAN3_Jiang Eq. (7)
where $y_d$ is a function of PFT.

MOFLUX_DroughtStress simulation with isoprene drought stress applied Eq. (8a-8c) is
found to reduce the MB at the MOFLUX site to $\cong$0.04 mg/m²/hr during the 2012 severe drought
period, indicating the parameterization is able to correct the model overestimation of isoprene
emissions. The NMB decreased to $\cong$1.53%, indicating a ~74.57% reduction compared to
Default_ModelE. Large improvements were not expected for 2011 as this algorithm was
designed to target severe/extreme drought. Despite the better agreement between measured and
modeled fluxes in MOFLUX_DroughtStress at the MOFLUX site, the regional analysis
described below determined that water stress values are region specific and a new approach was
needed in order to make the algorithm applicable for other regions in the global model.

**3.3 New Percentile Threshold Isoprene Drought Stress Parameterization**
After implementing MOFLUX_DroughtStress in ModelE, we found for JUN-AUG 2011
isoprene emissions reductions for the southeastern (SE) U.S. defined as (96-75°W, 25-38°N) of
approximately -3.5%, -7.2%, -5.7% respectively. These regional reductions were smaller than
expected as the SEUS 2011 was a spatially extensive severe drought over a largely forested and
vegetated region. The US Drought Monitor (USDM) reported that the southeast area in moderate
to exceptional drought for JUN-AUG 2011 was 63%, 61%, and 55% respectively. Other studies
for other regions of the world have reported during severe drought that reductions in isoprene
vary by region and have a large uncertainty. For example, Huang et al. (2015) reported using
different soil moisture products isoprene reductions of 12-70% for Texas. Others showed
reductions up to a maximum of 17% (Jiang *et al.* 2018; Wang *et al.* 2021). The reason why
MOFLUX_DroughtStress falls on the lowest end of reported isoprene reductions for the regional
analysis is probably because drought stress activation was calibrated to water stress ranges at a
single site. As water stress is expected to vary regionally, a new regional method was needed in
order to simulate drought stress effects globally.

A new parameterization was designed to not only work at MOFLUX since this is the site
used for validation, but capture isoprene drought signals for other regions. To do so, we first
simulated daily averaged water stress during the growing season for ten years (2003-2012) at
MOFLUX, a total of 2450 days. It was determined that water stress was less than the 0.4



threshold for 102 days, a percentage of ~ 4.16%. For simplicity, we rounded the percentage to
4%. The new approach then relied upon finding the 4$^{th}$ percentile water stress value across ten
years of daily water stress per grid and for each individual month in order to build a
parameterization that would capture regional and seasonal variability in water stress in ModelE.
This new drought stress parameterization is known as DroughtStress_ModelE and uses the same
alpha (α=100) as MOFLUX_DroughtStress and is applied as weighted average per PFT. What
makes this different from the previous approach, MOFLUX_DroughtStress, is that the water
stress threshold used to apply drought stress is based on the model's unique lowest 4$^{th}$ percentile
of water stress on a grid-by-grid basis and is not based on the absolute values of water stress at a
single site (i.e., MOFLUX). The 4$^{th}$ percentile of daily water stress was used as the trigger for
drought stress activation. The parameterization for **DroughtStress_ModelE** is Eq. (10a-10b):
$y_d = 1$   when ($\beta \geq$ 4$^{th}$ percentile)                                       (10a)
$y_d = \frac{(v_{c,max} \times \beta)}{\alpha}$   when  ($\beta <$ 4$^{th}$ percentile), where α=100       (10b)
A global transient simulation was run from (2003-2013) applying Eq. (10a-10b) globally,
called DroughtStress_ModelE in order to determine the effects of the isoprene drought stress
parameterization and to see if it captures the signal of the 2011 SE drought.
DroughtStress_ModelE for JJA 2011 showed isoprene emissions percent reductions for the SE of
approximately -9.6%, -5.9%, and -12.7% respectively. These reported reductions are a factor of
two greater than MOFLUX_DroughtStress for the same period, and are in the mid-range of
reported isoprene reductions during drought. A complete timeseries of isoprene emissions at
MOFLUX for all four simulations as described by **Table 1** is shown in SI Fig. S2a-b for 2011
and 2012.
**3.4 DroughtStress_ModelE Evaluation at MOFLUX**
During 2011 at the MOFLUX site, there were only small differences between
Default_ModelE and DroughtStress_ModelE. The scatterplots of isoprene emissions at the
MOFLUX site for the summer of 2011 show the hourly correlation coefficient between modeled
and observed isoprene fluxes showed minor improvement from 0.77 to 0.78, with minor changes
in slope and y-intercept (SI Fig. S3a,c). The diurnal cycles for 2011 included in (SI Fig. S4a)
showed that neither MOFLUX_DroughtStress nor DroughtStress_ModelE altered the diurnal
cycle in comparison to Default_ModelE. For 2011, all four simulations underestimate the diurnal
cycle for MAY-AUG. Large improvements due to the applications of the Eq. (10a-10b) were not
expected for 2011 as this algorithm was designed to target severe/extreme drought and not less
severe drought conditions.
During the severe drought period of 2012 at MOFLUX, the $\beta$ values fell below the 4$^{th}$
percentile thresholds for July-August, and isoprene drought stress was applied leading to
reductions in the overestimation shown by Default_ModelE. DroughtStress_ModelE had a MB





≅0.42 mg/m$^2$/hr and a NMB≅14.5%. DroughtStress_ModelE reduced overestimation by ~61.6%
compared to Default_ModelE, which is a similar statistical improvement compared to
MOFLUX_DroughtStress during the severe drought period as the parameterizations were
designed in a similar manner. The scatterplots of isoprene emissions at the MOFLUX site for the
summer of 2012 show the hourly correlation coefficient between observations and simulations
increased from 0.68 in Default_ModelE to 0.73 in DroughtStress_ModelE (Fig. 2a,c). In Fig. 2
changes are clearly seen in the cluster of $\beta$ values lower than 0.4 (shown by red oval) indicating
a reduction in overestimation during severe drought.

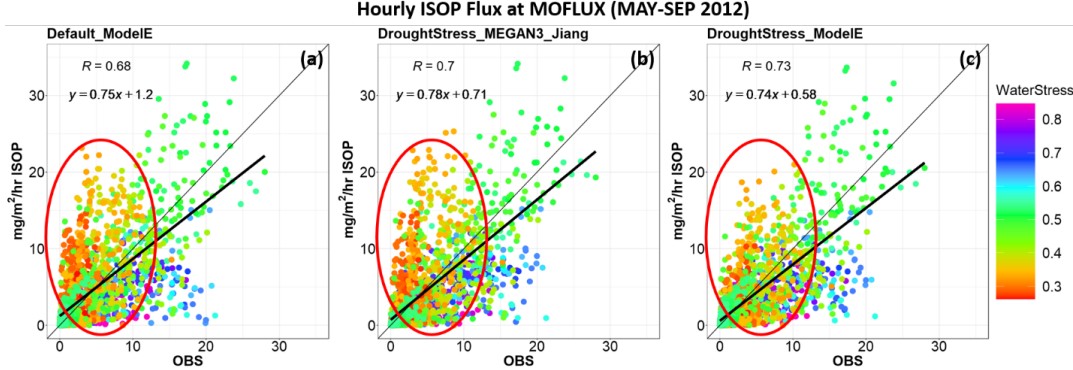

**Figure 2. Scatterplots (a-c) show hourly simulated isoprene emissions compared to observed for MAY-SEP 2012 at the**
**MOFLUX site and the units are mg/m$^2$/hr of isoprene. Column 1-3 indicate simulations Default_ModelE,**
**DroughtStress_MEGAN3_Jiang, and DroughtStress_ModelE respectively. The hourly averaged points are color coded by**
**water stress.**

DroughtStress_ModelE with decreases in y-intercept, increasing correlation coefficient, and
minor change in slope compared to Default_ModelE suggests it has better performance in
simulating isoprene emissions during severe and extreme drought at MOFLUX during the
summer of 2012. The daily correlation coefficient increased from 0.64 to 0.73 during severe
drought in DroughtStress_ModelE (SI Fig. S5a,c). In addition, DroughtStress_ModelE
reproduces the diurnal cycle of isoprene emission from MAY-SEP 2012 shown in (SI Fig. S4b*)*
and corrects the overestimation of the Default_ModelE during the peak hours 10-15 LST.
Overall, there is model agreement between measured and modeled fluxes in
DroughtStress_ModelE indicating it is a suitable model-tuned parameterization for estimating
isoprene emissions during severe drought at the MOFLUX site.

**4. Model response to drought parameterization: Global/Regional Evaluation of**
**DroughtStress_ModelE**
The impact of applying isoprene drought stress in DroughtStress_ModelE globally on the
annual emissions of isoprene from 2003-2013 is shown in **Table 2**. The yearly global reduction



of isoprene emissions ranges from ~ -0.9% to -4.3%. The global decadal average from 2003-
2013 is ~533 Tg yr$^{-1}$ of isoprene in Default_ModelE and ~518 Tg yr$^{-1}$ of isoprene in
DroughtStress_ModelE, a reduction of 2.7%, which is equivalent to ~14.6 Tg yr$^{-1}$ of isoprene.
On a global scale these changes average under 3%, but for high isoprene emission regions such
as the Southeast U.S. during drought periods there are larger impacts as shown below.

**Table 2. Global Annual Tg of Isoprene (2003-2013)**

| Global Annual Isoprene Emissions (Tg) | | | |
|---|---|---|---|
| Year | Default_ModelE | DroughtStress_ModelE | Diff (Tg Isoprene) | % Diff |
| **2003** | 557.5 | 533.4 | 24.1 | -4.3 |
| **2004** | 557.6 | 535.4 | 22.2 | -4.0 |
| **2005** | 578.6 | 562.1 | 16.5 | -2.9 |
| **2006** | 537.5 | 522.9 | 14.6 | -2.7 |
| **2007** | 527.2 | 515.8 | 11.4 | -2.2 |
| **2008** | 499.2 | 494.9 | 4.3 | -0.9 |
| **2009** | 522.3 | 508.4 | 13.9 | -2.7 |
| **2010** | 542.5 | 526 | 16.5 | -3.0 |
| **2011** | 508.3 | 498.8 | 9.5 | -1.9 |
| **2012** | 516.1 | 503.4 | 12.7 | -2.5 |
| **2013** | 512.5 | 497.5 | 15 | -2.9 |


Figure 3 shows the global nine-year average of isoprene emissions and tropospheric HCHO
column densities (ΩHCHO) of the lowest twenty layers of the model during JJA from 2005-
2013. Due to extremely limited in situ measurements of isoprene emissions during drought,
satellite-retrieved ΩHCHO, the high yield oxidation product of isoprene, can be used as a proxy
for isoprene emissions on the monthly scale (Zhu *et al.* 2016). Here we used ΩHCHO from OMI
(Ozone Monitoring Instrument) on the Aura satellite starting in 2005. Level 3 total column
weighted mean was regridded from its original resolution of 0.1°x0.1° to match ModelE's
horizontal resolution of 2°x2.5°, and the daily data was aggregated to monthly mean
([https://cmr.earthdata.nasa.gov/search/concepts/C1626121562-GES_DISC.html](https://cmr.earthdata.nasa.gov/search/concepts/C1626121562-GES_DISC.html)) (Chance 2019).
OMI satellite data was filtered with the data_quality_flag, cloud fractions less than 0.3, solar
zenith angles less than 60, and values within the range of $-0.5$ to $10 \times 10^{16}$ molecules cm$^{-2}$ were
used (Zhu *et al.* 2016). A factor of 1.59 is applied to the OMI vertical column density (VCD) to
correct the mean bias (Kaiser *et al.* 2018). Figures 3c,3f show the percent difference of isoprene
emissions and ΩHCHO and shown in blue are the decreases in DroughtStress_ModelE globally.
Figures 3d-e is OMI ΩHCHO and Default_ModelE simulated ΩHCHO. It is important to note
the difference in scales as Default_ModelE is overestimating ΩHCHO in regions such as the SE
U.S. for every June-July from the 2005-2013 period with a regional mean scale factor of ~0.56
and ~0.80 when the SE boundary is extended westward to include portions of Texas. These
overestimates in the SE U.S. are also reported by (Kaiser *et al.* 2018) where they saw a 50%





overestimate by GEOS-Chem with MEGAN2.1 simulations compared to SEAC[4]RS
observations. While applying isoprene drought stress leads to reductions in ΩHCHO as shown
by Fig. 3f, this reduction is limited to drought-stricken regions and periods and not designed to
correct for the systematic biases of HCHO in ModelE. The overestimation of ΩHCHO in
Default_ModelE will require further study and could be due to several reasons such as emissions
error, incorrect spatial gradient of OH, oxidation, or incorrect application of the sink of glyoxal
(Volkamer *et al.* 2007; Wells *et al.* 2020). This version of ModelE also lacks direct emissions of
HCHO from anthropogenic sources, which may result in the lower vertical deposition, and, due
to the short lifetime, the higher than observed HCHO column over portions of the U.S., and
lower in other regions.

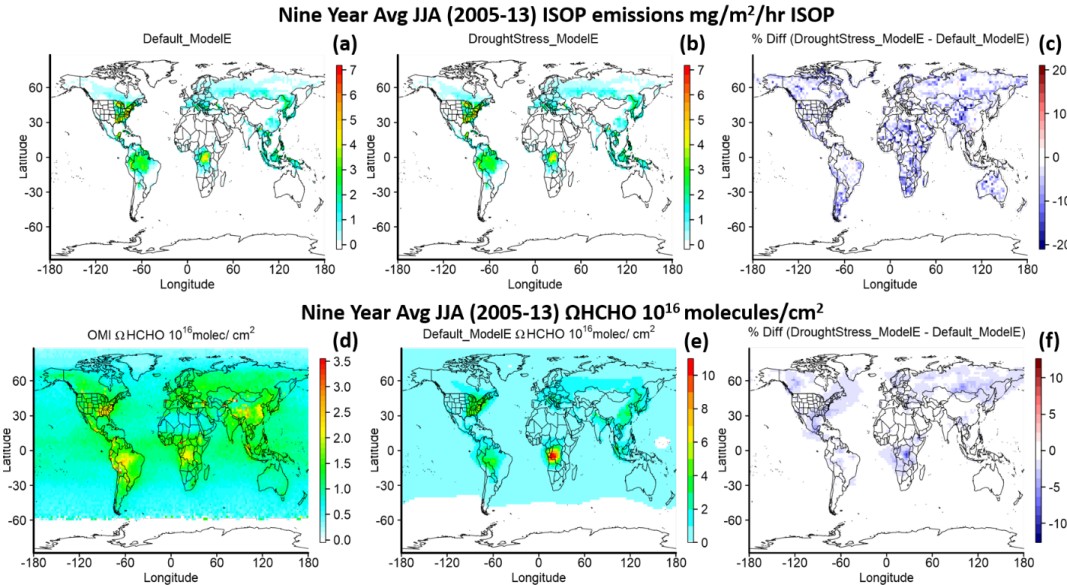

**Figure 3. Global nine-year average of JJA from 2005-2013 of isoprene emissions (first row) for Default_ModelE (a),**
**DroughtStress_ModelE (b) and percent difference between DroughtStress_ModelE and Default_ModelE (c), and**
**ΩHCHO (second row) for OMI (d), Default_ModelE (e) and percent difference between DroughtStress_ModelE and**
**Default_ModelE (f). Note the different color scales between (d) and (e).**

Four global isoprene emission hotspots are selected to showcase the changes in isoprene
emissions. The geographic regions are defined as East U.S. (Eastern U.S.: 65-105°W, 25-50°N),
SA (Amazon: 40-80°W, 30°S-7°N), AF (Central Africa: 10-40°E, 15°S-10°N), and SE Asia
(Southeast Asia: 100-150°E, 11°S-38°N) as shown in (SI Fig. S6). Figure 4 shows the
relationship of dryness categorized by SPEI (Standardized Precipitation-Evapotranspiration
Index) and relative difference in isoprene emissions between DroughtStress_ModelE and
Default_ModelE from 2005-2013 for the growing season in the northern hemisphere and
spring/summer in the southern hemisphere for the four global isoprene hotspots. SPEI is a



multiscalar climatic index that represents duration of drought in a region and is based on a
climatic water balance approach which considers the impact of temperature and
evapotranspiration (Beguería *et al.* 2010; Vicente-Serrano *et al.* 2010; Beguería *et al.* 2014). To
identify the extent of drought impacts and differentiate from normal variability in the
hydrological cycle, one-month SPEI is used to identify drought periods of duration extending
beyond a single month. Default_ModelE simulation variables were used to calculate modeled
SPEI at the resolution of 2°×2.5°. Positive SPEI typically indicates wet conditions and dry
conditions are indicated by negative values. Drought conditions are indicated by SPEI ≤ -1.3,
normal conditions -0.5 ≤ SPEI ≤ 0.5, and wet conditions SPEI ≥ 1.3 following the (Wang *et al.*
2017) approach. For the four regions the average percent difference in isoprene emissions for
March-October for northern hemisphere regions and September-February for southern
hemisphere regions from 2005-2013 is ~ -2.62% for the East U.S., the Amazon (SA) ~ -3.01%,
Central Africa (AF) ~ -2.64%, and Southeast Asia (SE Asia) ~ -3.10%. The scatterplots for the
four hotspots show decreasing isoprene emissions across all dryness conditions. The decreases in
isoprene emissions for the four regions are not seen exclusively when SPEI indicates dry
conditions, which indicates simulated water stress as shown by model does not align exactly with
SPEI drought indicated conditions.

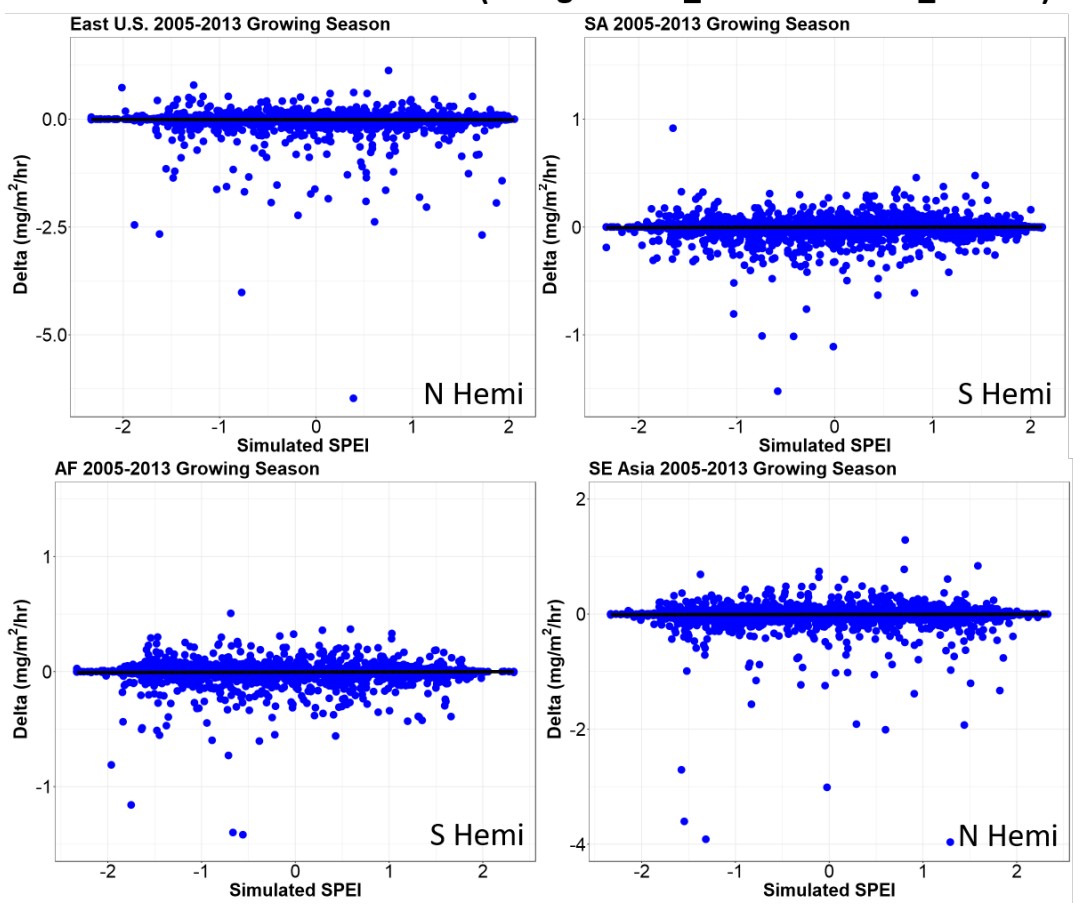

**Figure 4. The scatterplots of four global isoprene hotspot and their relative differences in isoprene emissions (mg/m$^2$/hr isoprene) in relationship to simulated SPEI from 2005-2013 during the growing season is shown. The four regions of focus are Eastern U.S. (East), Amazon (SA), Central Africa (AF), and Southeast Asia (SE Asia). The regions of East and SE Asia are in the northern hemisphere and the growing seasons is from (March-October). The hotspots of SA and AF are in the southern hemisphere and the growing season is during spring/summer (September-February).**

Narrowing the focus from global to the U.S., to illustrate the long-term difference between DroughtStress_ModelE and Default_ModelE, a timeseries from 2005-2013 is shown in Fig. 5 of the continental U.S. for two regions West (105-125°W, 25-50°N) and East (65-105°W, 25-50°N) indicating the percent difference in ΩHCHO and isoprene emissions corresponding to percent area that is dry (SPEI < -0.5). The map showing the regions West and East is located in (SI Fig. S7). The western U.S. (West) despite having a much smaller magnitude of isoprene emissions does see reductions in isoprene which is mimicked on a lesser scale by reductions in ΩHCHO. For the Eastern U.S. (East) there are clear decreases in isoprene emissions and ΩHCHO during



the droughts of 2007, 2011, and 2012. Focusing on the East timeseries, the maximum percent
difference between simulations DroughtStress_ModelE and Default_ModelE for isoprene
occurred from AUG-OCT 2007 approximately -4.5%, -7.4%, and -4.6% with corresponding
decreases in ΩHCHO of ~ -4.1%, -5.4%, and -3.6% respectively. For 2011 the maximum percent
difference in isoprene emissions occurred SEP-NOV and was ~ -9.0%, -8.7%, -8.3% and the
percent difference in ΩHCHO was ~ -5.9%, -3.6%, and -2.6%. For 2012 the maximum percent
difference occurred from AUG-OCT and the difference in isoprene was ~ -5.1%, -8.8%, and -
10.8% and the difference in ΩHCHO was ~ -2.8%, -4.0%, and -2.7%.





**Figure 5. The percent difference of ΩHCHO and isoprene emissions from 2005-2013 in relationship to percent area dry for two regions of the U.S. West (top figure) and East (bottom figure) is shown. Percent area dry is indicated by SPEI < - 0.5. The first grey shaded rectangle indicates the time period of the 2011 drought at MOFLUX from June to August 2011. The second grey shaded rectangle indicates the 2012 severe drought at MOFLUX from July 17 through August. These time periods are added to the timeseries to highlight when they occurred.**

Figure 6 displays spatial maps of ΩHCHO during the summer (JJA) of three drought years 2007, 2011, and 2012. The summers of 2007 and 2011 were drought periods in the U.S. with



2007 being a less severe drought than 2011 in the SE U.S. The drought of 2012 was focused
more on the Great Plains (GP) region. The spatial maps show the reduction in ΩHCHO in panels
6c, 6f, and 6i due to the inclusion of isoprene drought stress. Based on the spatial differences in
ΩHCHO, three regions of the greatest reduction in percent difference in ΩHCHO column are
selected for the three drought years of 2007, 2011, and 2012, respectively. The three geographic
regions are shown in Fig. 7 and defined as SE1(Southeast Region1: 75-93°W, 31-39°N), SE2
(Southeast Region2: 75-101°W, 29-37°N), and GP (Great Plains: 89-100°W, 33-43°N). During
JJA for 2007 the SE1 region has an average percent difference in ΩHCHO of -6.46%, during JJA
2011 the SE2 region has a percent difference of -7.58%, and the GP region during JJA 2012 has
average percent difference of -3.29%.

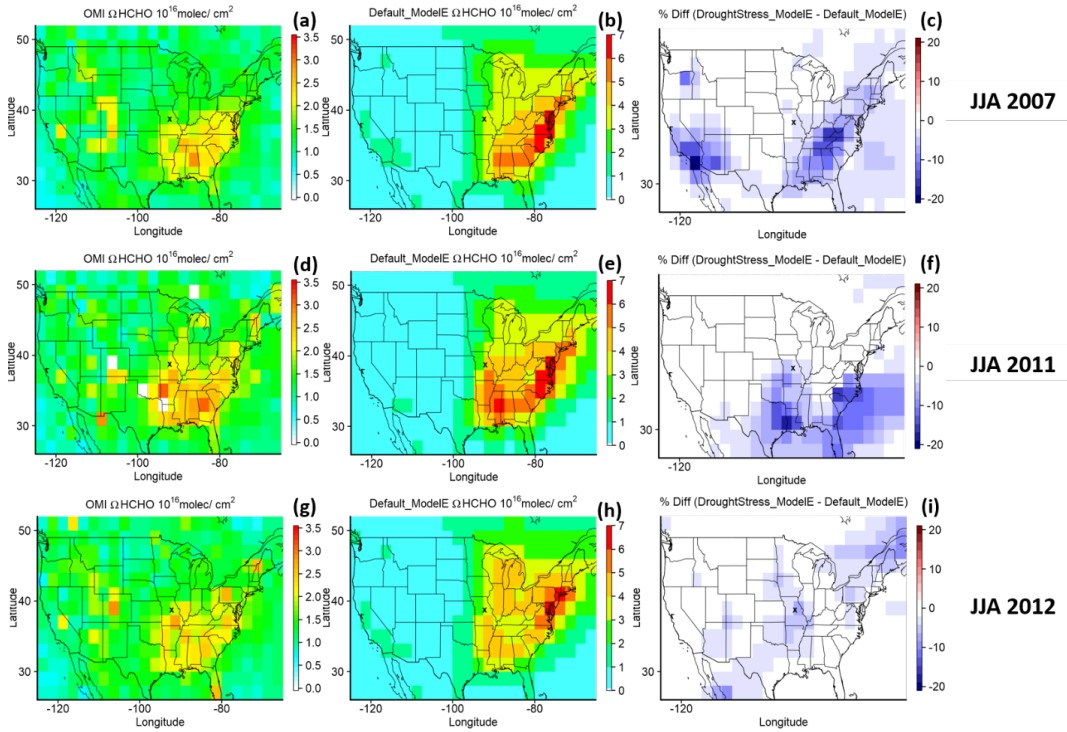

**Figure 6. The ΩHCHO column in units of molecules/cm² for OMI, Default_ModelE, and the percent difference between**
**DroughtStress_ModelE and Default_ModelE across the U.S. during the summer of drought years 2007, 2011, and 2012 is**
**shown. X indicates the location of the MOFLUX site on the spatial maps.**

Figure 7 shows the timeseries for the three regions of SE1 during 2007, SE2 for 2011, and
GP for 2012 drought. In the SE1 region during the period of maximum isoprene difference from
AUG-OCT 2007 shaded in grey on the timeseries, DroughtStress_ModelE reduced NMB of
ΩHCHO by ~19.3%. The isoprene percent difference for this period was approximately -9.0%, -
17.5%, and -13.2%. The ΩHCHO percent difference for the SE1 region from AUG-OCT 2007



was approximately -8.4%, -12.1%, and -7.3%. In the SE2 region the maximum isoprene
difference period for AUG-NOV 2011, DroughtStress_ModelE decreased $\Omega$HCHO NMB by
~15.3%. The monthly isoprene percent difference for SE2 during this period was approximately
-16.1%, -18.6%, -14.7%, and -13.9% while the $\Omega$HCHO percent difference was ~ -10.0%, -
11.2%, -6.6%, and -4.6% respectively. In the GP region during SEP-NOV 2012, the isoprene
percent difference for GP during SEP-NOV 2012 was approximately -5.4%, -14.2%, and -11.1%
and the $\Omega$HCHO percent difference was ~ -2.8%, -2.4%, and -0.4% respectively. The small
change in HCHO column despite estimated larger changes in isoprene emissions is probably due
to the suppression of oxidants such as hydroxyl radicals (OH) by isoprene under low-NOx
conditions in the GP region (Wells *et al.* 2020).
It is well established that biogenic isoprene, the most abundant BVOC, is a highly reactive
species. In the presence of nitrogen oxides ($NO_x$), BVOCs contribute to the formation of
tropospheric $O_3$. Oxidation of BVOCs also produces secondary organic aerosols, a major
component of fine particulate matter ($PM_{2.5}$). $PM_{2.5}$ and $O_3$ have been previously linked to
change during drought with adverse effects on air quality (Wang *et al.* 2017). It is thus important
to show the impact of drought-induced changes in isoprene emissions on $O_3$ and $PM_{2.5}$. The
scatterplots in Fig. 7 show the relationship between observed and simulated $O_3$ during the
drought period of maximum percent difference highlighted on the timeseries for the
corresponding region. $PM_{2.5}$ comparison to observed is not shown here due to Default_ModelE
underestimating $PM_{2.5}$ across all three regions SE1, SE2, and GP, and thus no improvements
were seen due to the inclusions of DroughtStress_ModelE. The observational $O_3$ data is a
combination of hourly data from the EPA-AQS (U.S. Environmental Protection Agency (EPA)
Air Quality System), CASTNET (Clean Air Status and Trends Network), and NAPS (National
Air Pollution Surveillance) networks. The observational $O_3$ datasets was gridded and interpolated
for comparison to a gridded model (Schnell *et al.* 2014). The hourly gridded observations were
then averaged onto a monthly scale for comparison with model results. Shown in Fig. 7 the SE1
region saw improvement in $O_3$ from AUG-OCT 2007, where the correlation coefficient (R)
increased from 0.51 in Default_ModelE to 0.60 in DroughtStress_ModelE and the slope of the
linear regression also improved significantly. The SE2 region from AUG-NOV 2011 saw a slight
improvement in the slope of the linear regression but no change in R. The GP region from SEP-
NOV 2012 saw a slight improvement in R but no change in the correlation slope between
Default_ModelE and DroughtStress_ModelE. During non-drought periods of 2008, 2010, and
2013 compared to their respective drought periods of 2007, 2011, and 2012 there was no large
changes in $O_3$ or $\Omega$HCHO statistics as expected since isoprene drought stress is only supposed to
effect drought periods. During the drought periods of 2007, 2011, and 2012 the model predicts
higher mean $O_3$ and $\Omega$HCHO than the non-drought years. The analysis of these drought years
and periods of the greatest percent difference leads to the conclusion of isoprene drought stress
improves $\Omega$HCHO simulation and $O_3$ simulation during drought periods.






**Figure 7. The timeseries from 2005-2013 of percent area dry on y-axis shown in red and percent difference in ΩHCHO**
**(blue) and isoprene emissions (black) between DroughtStress_ModelE and Default_ModelE for the 3 regions SE1, SE2,**
**and GP on the second y-axis is shown. Shaded in grey are the time periods of maximum percent difference of isoprene**
**emissions during the drought years. The scatterplots show the relationship between observed O₃ (ppbv) and simulated O₃**
**during the shaded grey time periods on the timeseries for Default_ModelE in black and DroughtStress_ModelE in red for**
**the SE1 during 2007, SE2 during 2011, and GP during 2012. Maps showing the geographic regions are inset into the**
**scatterplots. The regions spatial extent is based on region of maximum percent difference in Fig. 6c,f,i.**



### 5. Discussion and conclusions

Drought is a hydroclimatic extreme that causes perturbations to the terrestrial biosphere. As a stressor for vegetation, drought can induce changes to vegetative emissions known as BVOCs (Biogenic Volatile Organic Compounds). Biogenic isoprene represents about half of total BVOC emissions and is a precursor to ozone ($O_3$) and secondary organic aerosol (SOA), both of which are climate forcing species. In order to simulate isoprene flux during drought and the feedbacks associated with these complex BVOC-chemistry-climate interactions, we implemented the MEGAN (Model of Emissions of Gases and Aerosols from Nature) isoprene drought stress parameterization, $y_d$, into NASA GISS (Goddard Institute of Space Studies) ModelE, a leading Earth System Model. Four online transient simulations were performed from 2003-2013, a Default_ModelE without $y_d$, DroughtStress_MEGAN3_Jiang using the parameterization developed by (Jiang *et al.* 2018), and a model-tuned parameterization developed for ModelE based on the MOFLUX Ameriflux site observations (MOFLUX_DroughtStress). The fourth simulation implemented isoprene drought stress using a grid-by-grid approach to capture regional changes in isoprene during drought known as DroughtStress_ModelE. The model-tuned parameterization (MOFLUX_DroughtStress and DroughtStress_ModelE) was developed using an offline model of emissions to create a model specific empirical variable and water stress threshold, since key variables $V_{c,max}$ (photosynthetic parameter) and water stress ($\beta$) are parameterized differently across models. Observational measurements of isoprene flux during the severe drought of 2012 at the MOFLUX site were used for validation of parameterization. It was found that DroughtStress_ModelE corrects the overestimation of emissions during the phase of severe drought at MOFLUX. Previously, this reduction during drought was not included in BVOC emission models due to the lack of a drought stress term. Globally the decadal average from 2003-2013 in Default_ModelE was ~533 Tg of isoprene and ~518 Tg of isoprene in DroughtStress_ModelE. DroughtStress_ModelE was validated using observational satellite $\Omega$HCHO column from the Ozone Monitoring Instrument (OMI) and using $O_3$ observations across regions of the U.S. to examine the effect of drought on atmospheric composition. It was found that the inclusion of isoprene drought stress reduced the overestimation of $\Omega$HCHO in Default_ModelE during the 2007 and 2011 southeastern U.S. droughts and led to improvements in simulated $O_3$ during drought periods. The inclusions of a grid specific percentile isoprene drought stress is model specific and the reduction of isoprene seen in models will depend on each models mean bias and parameterizations of $V_{c,max}$ and water stress. ModelE's modest signal can be explained by underestimating isoprene emissions during the early stages of drought and by not having a high mean bias during severe drought.

Our analysis of isoprene drought stress leads to the recommendation that each model should arrive at a tuning of their water stress parameters based on the magnitude of water stress occurring during simulated drought and a unique alpha should be derived. Each land surface model (LSM) has a unique hydrology scheme (with different soil layering approaches and soil physics treatments), and any variables that depend on response to soil moisture -- whether



chemical, physical, or biological -- must be tuned due to the fact that soil moisture in LSMs is
being averaged over a grid cell whereas in nature soil moisture is heterogeneous at spatial scales
down to the plot level. The resulting parameterization, since it relies on model specific variables,
would be well suited for future or historical simulations. The current approach also requires
vegetation-coupled land surface models that have photosynthesis models that use $V_{c,max}$ and $\beta$,
and many current general circulation models (GCM) with less process-based vegetation schemes
do not have these variables readily available.
Besides tuning responses to drought, the light response of isoprene emissions may not be
well captured in a simple factor like the PCEEA. Vegetation models differ in their approach to
leaf-to-canopy scaling. Some ESMs vegetation models have more sophisticated canopy radiative
transfer submodels that capture layering and sunlit/shaded leaf area. Future isoprene modeling
investigations could make use of the ability of these canopy models to calculate isoprene
emissions with leaf-level responses to the heterogeneous light in canopies. Unger *et al.* (2013)
implemented such a leaf-to-canopy scaling of isoprene emissions previously in the Ent TBM
through a leaf-level isoprene model as a function of leaf-level gross primary production (GPP).
Since the Ent TBM scales stomatal conductance with drought stress, and hence also GPP, this
intrinsically results in isoprene emissions responsiveness to drought stress. The main challenge
will be to find consensus about the fundamental processed-based physics of isoprene emissions
at the leaf level. The method of Unger et al. (2013) was not used for this paper in order to
preserve the MEGAN3 features and test this particular isoprene drought stress parameterization.
A limitation of our tuning method for applying isoprene drought stress is that there does not
appear to be a strong relationship between SPEI and water stress, which makes it challenging to
determine when the algorithm should be applied during severe drought. This is why the current
application is limited and based on the single MOFLUX site where water stress values and the
corresponding decreases of isoprene during severe drought were observed. Possible future work
of the satellite Cross-track Infrared Sound (CrIS) isoprene measurements (Wells *et al.* 2020)
may be used to develop a drought algorithm that is not based on a single site and provide a more
dynamic drought stress algorithm for capturing the decrease of emissions during severe drought.
The reduction of isoprene in the model also depends on how dry (low values of water stress) the
model is. If the model is too dry or if isoprene emissions are already overestimated there will be
larger reductions in isoprene than reported here in ModelE, with larger feedbacks on $O_3$, SOA,
and $\Omega$HCHO column. Models that are not severely overestimating during severe drought will
show modest reductions like ModelE. It is important to note that the application of isoprene
drought stress in this paper is designed to reduce emissions during severe drought. Future work
could focus more on the parameterization of isoprene emissions during mild or early stages of
drought when isoprene emissions might be increasing and as we see in ModelE the model
underestimates during this period. Overall, the strength of the reduction signal of isoprene
depends on the model, and for models overestimating isoprene the application of isoprene



drought stress into the model could improve model simulations significantly. Recent published work has also brought up the importance of drought duration as an important factor to consider in further isoprene drought stress parameterization (Li *et al.* 2022). Future work on developing drought parameterizations should focus on capturing the increasing signal of isoprene at the start of drought, the reduction signal during severe drought, while also considering a time component because eventually plants can reach a stage of emission cessation.

In summary, this paper demonstrates why isoprene response to drought stress is model specific and should be tuned on a model-by-model basis, and details a new method for implementing isoprene drought stress to reduce isoprene emissions during severe drought in ModelE. This new method uses a grid-by-grid percentile threshold based on simulated water stress and can be used by many models to show regionals changes in isoprene emissions during severe drought and their associated feedbacks on $\Omega$HCHO and $O_3$. With more severe droughts predicted in the United States for the 21st century (Dai 2013), this is a first look into model performance for analyzing how BVOC emissions change during drought conditions using GISS ModelE for regions in the U.S.

## 6. Acknowledgements

E.K., Y.W. and A.G. would like to acknowledge the support and funding from the NASA ACMAP Program (80NSSC19K0986). E.K. and Y.W. would also like to acknowledge the support and funding of NASA Fellowship Grant (80NSSC18K1704) and thank support of NASA technical advisors at Goddard Institute of Space Studies. Resources supporting this work were provided by the NASA High-End Computing (HEC) Program through the NASA Center for Climate Simulation (NCCS) at the Goddard Space Flight Center. GISS authors acknowledge funding from the NASA Modeling and Analysis program.

## 7. Data availability

ModelE is publicly available at https://simplex.giss.nasa.gov/snapshots/ and $O_3$ and $PM_{2.5}$ observational data available for download via https://aqs.epa.gov/aqsweb/documents/data_mart_welcome.html. Observational isoprene measurements at MOFLUX are from Potosnak *et al.* 2014 and Seco *et al.* 2015 and are available upon request from co-author Alex Guenther. MOFLUX is part of the Ameriflux network and other observational data is available for download at https://ameriflux.lbl.gov/sites/siteinfo/US-MOz#BADM. Satellite $\Omega$HCHO is available publicly at https://cmr.earthdata.nasa.gov/search/concepts/C1626121562-GES_DISC.html.

## 8. Author contribution

EK and YW conceived the research idea. EK wrote the initial draft, conducted the simulations, and performed the analysis. EK and GF conducted model development. All authors contributed to the interpretation of the results and the preparation of the paper.

## 9. Competing interests

The authors declare that they have no conflict of interest.



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
