# Peer review of "Interactive Biogenic Emissions and Drought Stress Effects on Atmospheric Composition in NASA GISS ModelE"

_EGUsphere, 2022_

## Referee Comment (RC2)

**Review of Klovenski et al. 2022**

Klovenski et al. present research investigating the impact of drought stress on isoprene emissions and atmospheric composition in the NASA GISS ModelE Earth system model. They implement the drought stress parameterization from the MEGAN3 model, and apply a modelspecific tuning method to best reproduce observed isoprene fluxes at the Missouri. They then compare the results of their simulated drought impacts on atmospheric composition with observed formaldehyde and ozone concentrations. While the work is generally well written and addresses an important research topic, I cannot recommend the manuscript for publication in its current form. Major and minor comments affecting this recommendation are summarized below.

**Major Comments**

**MOFLUX Point Comparisons** – Much of this work is based around tuning the emissions scheme to observations at one location: the MOFLUX field site. Tuning a global model to one individual site is suboptimal, but necessary in this case given the limited data available on isoprene emissions and drought. However, more work should be done demonstrating that this tuning is not compensating for substantial model errors that lead to differences in prediction. At minimum in order to assess the validity of this tuning factor, it would be useful to see the model performance on other variables necessary for predicting isoprene emissions. Example pertinent questions include:

Does the model properly simulate meteorological drivers of emissions at the MOFLUX site?

Does the land classification in the model match the observed site? Does the model properly represent vegetation properties (e.g., LAI, PFT, etc.)? How do the differences in simulated emissions and observations compare with the substantial uncertainty estimates in the MEGAN model (Guenther et al., 2012)?

The scatterplot in Figure 2 and associated discussion shows a very limited improvement in R2 (0.03). Since this work is entirely focused on water stress, it would be useful to have those metrics only for the water stressed time periods.

**Model Tuning**

The authors recommend applying the drought stress when the model grid cell water stress is in below the 4th percentile. I understand the logic for this choice, but it should be contextualized further. Is there reason to expect that vegetation respond to relative or absolute water stress? Is this tuning representative of any physical or biological process, or simply statistical?

**Formaldehyde Comparisons** – The analysis of formaldehyde retrievals in this work may be lacking relative to the state-of-the-science and does not support the conclusions in the manuscript.

For apples-to-apples comparisons between satellite observations of formaldehyde and models, the Air Mass Factor (AMF) should be recalculated and applied to the observations. That was not

done in this work. At the very least that point and the associated limitations imposed should be discussed.

The ModelE simulated formaldehyde column disagrees substantially with observations (e.g., Figure 6). It appears as though the column is overestimated by at least a factor of 3. This enormous overestimation is not common across other models of atmospheric chemistry (e.g., GEOS-Chem), and calls into question the validity of ModelE simulated formaldehyde concentrations. While adding the drought stress does improve the simulation, that result alone is not interesting as anything that reduces formaldehyde concentrations would improve the simulation. The authors should make a stronger case as to why the ModelE formaldehyde simulations should be trusted as a useful assessment tool for isoprene emissions changes.

**Statistical significance of results**

Many of the results here are lacking detailed statistical treatment to understand if the results are either statistically or practically useful, in particular Figure 2, Figure 4, Figure 5, and Figure 7. All these figures and associated discussion describe noisy results. I am sympathetic to the challenges related to non-drought variability in constraining the process the authors are addressing, but substantially more rigorous assessment is needed before these results can be assessed in depth.

For example, the trends shown in Figure 4 do not appear to be significant in any way. They visually look to be a random sampling of points scattered about y = 0, and do not appear to show "decreasing isoprene emissions" as claimed in the text.

**Minor Comments**

The introduction includes substantive discussion of drought impacts on SOA, but the analysis does not assess SOA at all, this is confusing.

L548: How was the selection of an alpha value of 100 made? "Best fit" by what metric?

L657: "there is model agreement" is far too strong of a statement given the large scatter in Figure 2.

L695-698: This sentence around HCHO overestimation is confusing. Are there any reasons why the authors suspect these reasons are the culprit for overestimation? If so, why were they not addressed in more detail during model development?

L747: "clear decreases" are not evident given the substantial variability in the figure. See discussion of statistical significance above.

L823: These changes in HCHO and O3 are very small relative to the large biases that still exist, particularly in rows 2 and 3 of Figure 7.

---

## Author Comment (AC1)

**Reply to Reviewer 1**

We sincerely appreciate the two reviewers for their constructive comments to improve the manuscript. Reviewer 1's comments are reproduced below with our responses in blue. The corresponding edits in the manuscript are highlighted with track changes.

**Specific Comments**

Line 74. The CO2 parameterisation serves to inhibit isoprene emissions (Heald et al., 2009)? **Response:** Thank you for the comment about the need to expand further on this topic. In MEGAN2.1 there is a parameterization used called  $y_{co_2}$  which inhibits and decreases nonlinearly isoprene emissions when CO2 concentration rises above 400ppmv which is included in our MEGAN implementation in NASA GISS ModelE following (Heald *et al.* 2009; Guenther *et al.* 2012; Henrot *et al.* 2017) as described in line 182-183.

Please see line 71-75 for statement that clarifies our intent "Biogenic isoprene emissions affect atmospheric composition and climate, and in turn depend on environmental factors including light, temperature, photosynthetically active radiation (PAR), leaf area index (LAI), water stress, ambient O3, and CO2 concentrations. Thus, the response of isoprene emissions to weather extremes and changing climates is highly uncertain."

Line 80-82. Very confusing sentence. 'During drought, increases in SOA and O3 are to be expected'. Why? What aspect of drought would cause this? (In my mind, less isoprene would mean less ozone?) Needs explanation. Then the second part of the sentence suggests isoprene reductions will decrease the magnitude of the increase.

**Response:** We agree that the message is not clearly conveyed and our discussion of SOA in relation to our study is inappropriate for the introduction, thus we have removed mentions of SOA in accordance with a suggestion from Reviewer 2.

We've expanded the explanation of ozone behavior during drought on line 882-889. "During drought there is elevated  $O_3$  and  $PM_{2.5}$ , compared to non-drought periods (Wang *et al.* 2017; Zhao *et al.* 2019; Naimark et al., 2021). Higher ozone compared to non-drought years is due to the reduction of vegetative deposition due to reduced stomatal conductance, higher temperatures stimulating precursors, and enhanced NO2 (Naimark *et al.* 2021). By including isoprene drought stress into the simulations, isoprene emissions are decreased which will change  $O_3$ , and the direction of change depends on NOX-limited or VOC-limited regimes (Li *et al.* 2022). In summary, we better predicted isoprene emission response to drought by including isoprene drought stress."

Line 199. 1x10-9/3600 looks like it also contains the conversion from the emission factor units of ug to kg (rather than just being a timestep conversion). **Response:** Thank you for catching this typo. It is fixed on line 213-215. " $(1x10^{-9}/3600)$ : the numerator converts units from ug/m2/hr to kg/m2/s and the denominator is the timestep conversion for seconds in an hour."

Line 427 define USDM

**Response:** On line 402-407 we included the explanation of USDM. "The U.S. Drought Monitor (USDM) produces color-coded maps indicating drought severity across the U.S. and is produced through a partnership of the National Drought Mitigation Center at the University of Nebraska-Lincoln, the U.S. Department of Agriculture, and the National Ocean and Atmospheric Administration (NOAA). The USDM drought maps have five classifications to indicate drought condition: (D0) indicating abnormally dry, (D1) moderate drought, (D3) extreme drought, and (D4) exceptional drought."

Figure 1. there looks like a gap in the observations towards the end of the timeseries (mid august). Consider whether these time periods should be removed? **Response:** You are correct there was an equipment failure in mid-August 2011 at the MOFLUX site, which is documented in Potosnak *et al.* 2014. For the timeseries and analysis shown when there was missing data, we removed the corresponding simulated data to match only the periods with observations.

Lines 511-520. You talk about the general over/under estimation of the model but what about the shape of the fit details? Does the model hit or miss the daily peaks? What could be the reason for the missed peaks?

**Response:** The shape of the fit for the MAXVOC, severe drought, and drought recovery period is shown as the distribution of daily averaged values in **Fig. R1** shown below. During the MAXVOC period the means for Default\_ModelE and DroughtStress\_ModelE are below observed shown by yellow diamonds. During the severe drought period DroughtStress\_ModelE shown in green has a closer mean to observed shown in black indicating reduced emissions. During the drought recovery period there is little change in the distribution between Default\_ModelE and DroughtStress\_ModelE. In Fig. R2 are shown the time series of daily peaks in the observations, Default\_ModelE and DroughtStress\_ModelE. During the severe drought period, DroughtStress\_ModelE still is biased higher than the observed daily peak, but, except for one day (2012/08/01), consistently less than Default\_ModelE.

**Figure R1**: (a) boxplots to indicate the distribution of daily averaged isoprene emissions for the three simulations Default\_ModelE shown in red, DroughtStress\_ModelE shown in green, and observations show in black. (b) the distribution of isoprene during the severe drought and (c) the distribution during the drought recovery period with the averages shown by yellow diamond.

**Figure R2** shown below is the timeseries of hourly peak isoprene for each day for the time period MAY-SEP 2012. Default\_ModelE tends to underestimate the hourly peak of each day in the MAXVOC period. Default\_ModelE for much of severe drought period is higher than observed compared to observed hourly peak for each day. DroughtStress\_ModelE in green tends to reduce the daily peak and move it closer to observed during severe drought period. During drought recovery there is not much difference between Default\_ModelE and DroughtStress\_ModelE daily peaks.

---

## Author Comment (AC2)

**Reply to Reviewer 2**

We sincerely appreciate the two reviewers for their constructive comments to improve the manuscript. Reviewer 2's comments are reproduced below with our responses in blue. The corresponding edits in the manuscript are highlighted with track changes.

**Major Comments**

**MOFLUX Point Comparisons** – Much of this work is based around tuning the emissions scheme to observations at one location: the MOFLUX field site. Tuning a global model to one individual site is suboptimal, but necessary in this case given the limited data available on isoprene emissions and drought. However, more work should be done demonstrating that this tuning is not compensating for substantial model errors that lead to differences in prediction.

At minimum in order to assess the validity of this tuning factor, it would be useful to see the model performance on other variables necessary for predicting isoprene emissions. Example pertinent questions include:

Does the model properly simulate meteorological drivers of emissions at the MOFLUX site?

**Response**: Temperature is the main driver of biogenic isoprene (Mishra and Sinha 2020; Jiang *et al.* 2018). In response to the first reviewer, we included the timeseries showing daily averaged temperature compared to observed at the MOFLUX site during MAY-SEP 2012 in the updated supplement as **Fig. S10** and is included below as **Fig. R1**. ModelE does a reasonable job reproducing the temperature at the site which gives us confidence the meteorological drivers of biogenic isoprene are correct. We used nudged NCEP meteorology to simulate 2003-2013 and any changes in the meteorology are due to interactions in the model.

**Figure R1**: shows the timeseries of daily averaged (LST) temperature at MOFLUX site for MAY-SEP 2012 in Celsius. The observed temperature is shown in black and red shows Default\_ModelE.

Gu *et al.* (2006) detail how the exchange of latent and sensible heat fluxes is one of the most important aspects of land-atmosphere coupling as these energy fluxes are affected by partitioning

of net radiation absorbed by the surface, which influence atmospheric dynamics, influence boundary layer structure, cloud development, and rainfall. Thus, we verified latent heat and sensible heat at the MOFLUX site and compared observed to simulated during MAY-SEP 2012, which we have included in the revised supplement as **Fig. S11**. We found from MAY-SEP 2012 Default\_ModelE does a reasonable job reproducing hourly sensible heat with a correlation coefficient (R) of 0.83 and slope of 1. For MAY-SEP 2012, Default\_ModelE has a R of 0.60 and slope of 0.52 when comparing to observed hourly averaged latent heat as shown below in **Fig. R2**.

---

## Author Response (AR2)

On page 3 line 122 -124, is this sentence correct (it is the sentence that starts with "The model drought's effects..."? This sentence makes it sound like you will evaluate your model again many measurements during drought in the US, but the reference is to a paper in China and the next sentence states how few measurements there are in the US. So I find it a little confusing. On line-122-130 we clarify our intent as this was unintentionally a confusing statement. We meant to convey that very few direct measurements of isoprene during drought exist except for the MOFLUX site and for our paper we focus on the U.S. region and use the EPA AQS monitoring network such as was used in a prior study by the the reference (Wang *et al.* 2017). We use satellite formaldehyde column data to fill in gaps since it's a proxy for isoprene emissions with the reference from (Zhu *et al.* 2016).

"The model's drought effects were extensively evaluated across the U.S., due to the availability of observational evidence during drought at the MOFLUX site and due to the EPA monitoring network for $O_3$ and $PM_{2.5}$ (Wang *et al.* 2017). While the MOFLUX data are the only available measurements of isoprene emissions during drought, formaldehyde (HCHO), the high yield oxidation product of isoprene, can be used as a proxy for isoprene emissions when no direct ground based observations of isoprene are available (Zhu *et al.* 2016)."

In figure 1a and figure S2a there are missing data I believe in mid-Aug (as noted in your response to reviewer 1. However, there isn't a gap in the timeseries. But rather it seems like there is just a line from the last point before the gap to the first point after the gap. I find this a bit confusing as that line could be taken for data. I would recommend leaving that time period as a gap rather. There is now a gap as requested for the missing data period of 2011 in both the paper and the supplement.

In sections 3.3, 3.4 and 4 there are sometimes when the months are in all caps (e.g. MAY-AUG). Please update those to match the style for writing months in rest of the paper (e.g. May-Aug). This has now been fixed in the paper sections listed above and in the supplement.

In table 2 for 2010, 526 is the only value without a tenths place. I would recommend keeping the same number of significant figures and updating this to 526.x (whatever "x" is). This is now fixed, as data was rounded to the tenths decimal place. The value 525.9765 is rounded to ~526.0.